# DROID-SLAM: Deep Visual SLAM for Monocular, Stereo, and RGB-D Cameras

**Zachary Teed**    **Jia Deng**
Princeton University
{zteed,jiadeng}@princeton.edu

## Abstract

We introduce DROID-SLAM, a new deep learning based SLAM system. DROID-SLAM consists of recurrent iterative updates of camera pose and pixelwise depth through a Dense Bundle Adjustment layer. DROID-SLAM is accurate, achieving large improvements over prior work, and robust, suffering from substantially fewer catastrophic failures. Despite training on monocular video, it can leverage stereo or RGB-D video to achieve improved performance at test time. The URL to our open source code is https://github.com/princeton-vl/DROID-SLAM.

## 1 Introduction

Simultaneous Localization and Mapping (SLAM) aims to (1) build a map of the environment and (2) localize the agent within the environment. It is a special form of Structure-from-Motion (SfM) focused on accurate tracking of long-term trajectories. It is a critical capability for robotics, especially autonomous vehicles. In this work, we address *visual* SLAM, where sensor recordings come in the form of images captured from a monocular, stereo, or RGB-D camera.

The SLAM problem has been approached from a number of different angles. Early work was built using probabilistic and filtering based approaches [12, 30], and alternating optimization of the map and camera poses [34, 16]. More recently, modern SLAM systems have leveraged least-squares optimization. A key element for accuracy has been full Bundle Adjustment (BA), which jointly optimizes the camera poses and the 3D map in a single optimization problem. One advantage of the optimization-based formulation is that a SLAM system can be easily modified to leverage different sensors. For example, ORB-SLAM3 [5] supports monocular, stereo, RGB-D, and IMU sensors, and modern systems can support a variety of camera models [5, 27, 42, 6]. Despite significant progress, current SLAM systems lack the robustness demanded for many real-world applications. Failures come in many forms, such as lost feature tracks, divergence in the optimization algorithm, and accumulation of drift.

Deep learning has been proposed as a solution to many of these failure cases. Previous work has investigated replacing hand-crafted with learned features[13, 7, 29, 26, 35], using neural 3D representations[46, 1, 9, 45, 44, 25, 22], and combining learned energy terms with classical optimization backends[58, 57]. Other work has tried to learn SLAM or VO systems end-to-end [59, 47, 53, 52, 46]. While these systems are sometimes more robust, they fall far short of the accuracy of their classical counterparts on common benchmarks.

In this work we introduce DROID-SLAM, a new SLAM system based on deep learning. It has state-of-the-art performance, outperforming existing SLAM systems, classical or learning-based, on challenging benchmarks with very large margins. In particular, it has the following advantages:

- *High Accuracy*: We achieve large improvements over prior work across multiple datasets and modalities. On the TartanAir SLAM competition [54], we reduce error by 62% over the best prior result on the monocular track and 60% on the stereo track. We rank 1st on the

35th Conference on Neural Information Processing Systems (NeurIPS 2021).

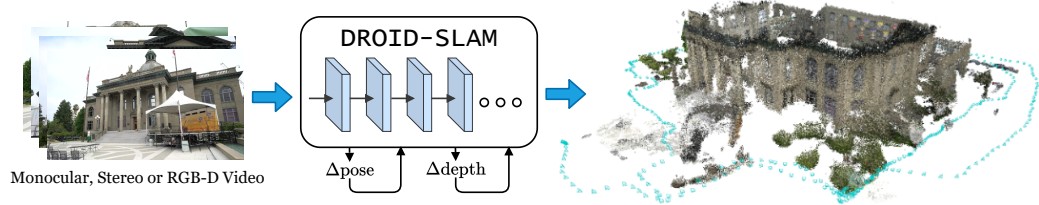

Figure 1: DROID-SLAM can operate on monocular, stereo, and RGB-D video. It builds a dense 3D map of the environment while simultaneously localizing the camera within the map.

ETH-3D RGB-D SLAM leaderboard [41], outperforming the second place by 35% under the AUC metric which considers both error and rate of catastrophic failure. On EuRoC [2], with monocular input, we reduce error by 82% among methods with zero failures, and by 43% over ORB-SLAM3 considering only the 10 out of 11 sequences it succeeds on. With stereo input, we reduce error by 71% over ORB-SLAM3. On TUM-RGBD [43], we reduce error by 83% among the methods with zero failures.

- *High Robustness*: We have substantially fewer catastrophic failures than prior systems. On ETH-3D, we successfully track 30 of the 32 RGB-D datasets, while the next best successfully tracks only 19/32. On TartanAir, EuRoC, and TUM-RGBD, we have zero failures.

- *Strong Generalization*: Our system, trained only with monocular input, can directly use stereo or RGB-D input to get improved accuracy without any retraining. All of our results across 4 datasets and 3 modalities are achieved by a single model, trained once with only monocular input entirely on the synthetic TartanAir dataset.

The strong performance and generalization of DROID-SLAM is made possible by its "Differentiable Recurrent Optimization-Inspired Design" (DROID), which is an end-to-end differentiable architecture that combines the strengths of both classical approaches and deep networks. Specifically, it consists of recurrent iterative updates, building upon RAFT [48] for optical flow but introducing two key innovations.

First, unlike RAFT, which iteratively updates optical flow, we iteratively update camera poses and depth. Whereas RAFT operates on two frames, our updates are applied to an arbitrary number of frames, enabling joint global refinement of all camera poses and depth maps, essential for minimizing drift for long trajectories and loop closures.

Second, each update of camera poses and depth maps in DROID-SLAM is produced by a differentiable Dense Bundle Adjustment (DBA) layer, which computes a Gauss-Newton update to camera poses and *dense per-pixel depth* so as to maximize their compatibility with the current estimate of optical flow. This DBA layer leverages geometric constraints, improves accuracy and robustness, and enables a monocular system to handle stereo or RGB-D input without retraining.

The design of DROID-SLAM is novel. The closest prior deep architectures are DeepV2D [47] and BA-Net [46], both of which were focused on depth estimation and reported limited SLAM results. DeepV2D alternates between updating depth and updating camera poses, instead of bundle adjustment. BA-Net has a bundle adjustment layer, but their layer is substantially different: it is not "dense" in that it optimizes over a small number of coefficients used to linearly combine a depth basis (a set of pre-predicted depth maps), whereas we optimize over per-pixel depth directly, without being handicapped by a depth basis. In addition, BA-Net optimizes photometric reprojection error (in feature space), whereas we optimize geometric error, leveraging state-of-the-art flow estimation.

We perform extensive evaluation across four different datasets and three different sensor modalities, demonstrating state-of-the-art performance in all cases. We also include ablation studies that shed light on important design decisions and hyperparameters.

## 2 Related Work

Modern SLAM systems treat localization and mapping as a joint optimization problem [4].

**Visual SLAM** focuses on observations in the form of monocular, stereo, or RGB-D images. These approaches are commonly categorized as either being *direct* or *indirect* [15]. Indirect approaches [31, 32, 5, 37] first process the image into an intermediate representation by detecting points of interest and attaching feature descriptors. Features are then matched between images. Indirect approaches optimize camera pose and a 3D point cloud by minimizing reprojection error–the distance between a projected 3D point and its location in the image.

Direct approaches model the image formation process and define an objective function over photometric error [16, 15, 60]. One advantage of direct approaches is that they can model more information about the image, such as lines and intensity variations[15] which are not used by indirect approaches. However, photometric errors typically lead to more difficult optimization problems, and direct approaches are less robust to geometric distortion such as rolling shutter artifacts. This approach requires more sophisticated optimization techniques, such as coarse-to-fine image pyramids to avoid local minimum.

Our method does not clearly fit into either of the categories. Like the *direct* approach, we do not require preprocessing steps to detect and match features between the images. We instead use the full image, allowing us to leverage a wider range of information than indirect methods with typically only use corners and edges. However, we minimize reprojection error similar to indirect methods. This is an easier optimization problem and avoids the need for more complicated representations such as image pyramids. In this sense, our approach borrows the best of both approaches: the smoother objective function of indirect approaches with the greater modeling capacity of indirect approaches.

**Deep Learning** has more recently been applied to the SLAM problem. Many works have focused on training systems for particular subproblems, such as feature detection [13, 7, 29, 26, 35], feature matching and outlier rejection [38, 36], and localization [51, 39]. SuperGlue [38] was designed to perform feature matching and verification and make 2-view pose estimate much more robust. Our network also draws inspiration from Dusmanu et al[14], which builds a neural network into the SfM pipeline to improve keypoint localization accuracy.

Other works have focused on training SLAM systems end-to-end [59, 46, 8, 50, 24, 47, 53]. These methods are not full SLAM systems, but instead focus on small scale reconstruction on the order of two [8, 50, 53] up to a dozen frames [59, 46, 47]. They lack many of the core capabilities of modern SLAM systems such as loop closure and global bundle adjustment which inhibit their ability to perform large scale reconstruction as demonstrated in our experiments. $\nabla$SLAM[23] implements several existing SLAM algorithms as differentiable computation graphs, allowing for errors in the reconstruction to be backpropagated back to sensor measurements. While this approach is differentiable, it has no trainable parameters, meaning the performance of the system is limited by the accuracy of the classical algorithm they emulate.

DeepFactors[9] is the most complete deep SLAM system, building on the earlier CodeSLAM[1]. It performs joint optimization of the pose and depth variables, and is capable of short and long-range loop closure. Similar to BA-Net[46], DeepFactors optimizes the parameters of a learned depth basis during inference. In contrast, we do not rely on a learned basis, but instead optimize pixelwise depth. This allows our network to better generalize to new datasets since our depth representation is not tied to the training dataset.

## 3   Approach

We take a video as input with two objectives: estimate the trajectory of the camera and build a 3D map of the environment. We first describe the monocular setting; in Sec. 3.4 we describe how to generalize the system to stereo and RGB-D video.

**Representation:** Our network operates on an ordered collection of images, $\{\mathbf{I}_t\}_{t=0}^N$. For each image $t$, we maintain two state variables: camera pose $\mathbf{G}_t \in SE(3)$ and inverse depth $\mathbf{d}_t \in \mathbb{R}_+^{H \times W}$. The set of poses, $\{\mathbf{G}_t\}_{t=0}^N$, and set of inverse depths $\{\mathbf{d}_t\}_{t=0}^N$ are unknown state variables, which get iteratively updated during inference as new frames are processed. For the reminder of the paper, when we refer to depths, note that we are using the inverse depth parameterization.

We adopt a frame-graph $(\mathcal{V}, \mathcal{E})$ to represent co-visibility between frames. An edge $(i, j) \in \mathcal{E}$ means image $\mathbf{I}_i$ and $\mathbf{I}_j$ have overlapping fields of view which shared points. The frame graph is built dynamically during training and inference. After each pose or depth update, we can recompute

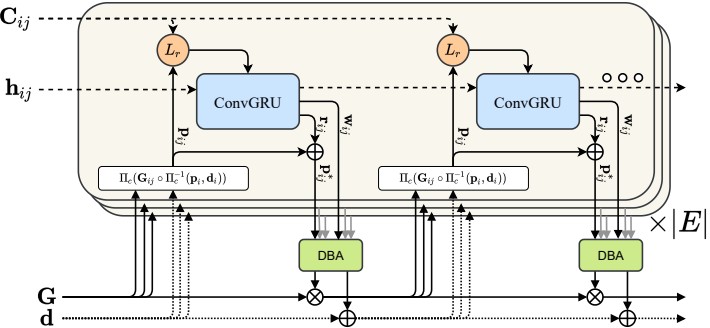

Figure 2: Illustration of the update operator. The operator acts on edges in the frame graph, predicting flow revisions which are mapped to depth and pose update through the (DBA) layer.

visibility to update the frame graph. If the camera returns to a previously mapped region, we add long range connections in the graph to perform loop closure.

### 3.1 Feature Extraction and Correlation

Features are extracted from each new image added to they system. Key components of this stage are borrowed from RAFT[48].

**Feature Extraction** Each of the input images are processed by a feature extraction network. The network consists of 6 residual blocks and 3 downsampling layers, producing dense feature maps at 1/8 the input image resolution. Like RAFT[48], we use two separate networks: a feature network and a context network. The feature network is used to build the set of correlation volumes, while the context features are injected into the network during each application of the update operator.

**Correlation Pyramid** For each edge in the frame graph, $(i, j) \in \mathcal{E}$, we compute a 4D correlation volume by taking the dot product between all-pairs of feature vectors in $g_\theta(I_i)$ and $g_\theta(I_j)$

$$C^{ij}_{u_1 v_1 u_2 v_2} = \langle g_\theta(I_i)_{u_1 v_1}, g_\theta(I_j)_{u_2 v_2} \rangle \tag{1}$$

We then perform average pooling of the last two dimension of the correlation volume following RAFT[48] to form a 4-level correlation pyramid.

**Correlation Lookup** We define a lookup operator which indexes the correlation volume using a grid with radius $r$, $L_r : \mathbb{R}^{H \times W \times H \times W} \times \mathbb{R}^{H \times W \times 2} \mapsto \mathbb{R}^{H \times W \times (r+1)^2}$.

The lookup operator takes an $H \times W$ grid of coordinates as input and values are retrieved from the correlation volume using bilinear interpolation. The operator is applied to each correlation volume in the pyramid and the final feature vector is computed by concatenating the results at each level.

### 3.2 Update Operator

The core component of our SLAM system is a learned *update operator* show in Fig. 2. The update operator is a $3 \times 3$ convolutional GRU with hidden state $\mathbf{h}$. Each application of the operator updates the hidden state, and additionally produces a pose update, $\Delta \boldsymbol{\xi}^{(k)}$, and depth update, $\Delta \mathbf{d}^{(k)}$. The pose and depth updates are applied to the current depth and pose estimates through retraction on the SE3 manifold and vector addition respectively

$$\mathbf{G}^{(k+1)} = \text{Exp}(\Delta \boldsymbol{\xi}^{(k)}) \circ \mathbf{G}^{(k)}, \qquad \mathbf{d}^{(k+1)} = \Delta \mathbf{d}^{(k)} + \mathbf{d}^{(k)}. \tag{2}$$

Iterative applications of the update operator produce a sequence of poses and depths, with the expectation of converging to a fixed point $\{\mathbf{G}^{(k)}\} \rightarrow \mathbf{G}^*, \{\mathbf{d}^{(k)}\} \rightarrow \mathbf{d}^*$, reflecting the true reconstruction.

**Correspondence** At the start of each iteration we use the current estimates of poses and depths to estimate correspondence. Given a grid of pixel coordinates, $\mathbf{p}_i \in \mathbb{R}^{H \times W \times 2}$ in frame $i$, we compute the dense correspondence field $\mathbf{p}_{ij}$

$$\mathbf{p}_{ij} = \Pi_c(\mathbf{G}_{ij} \circ \Pi_c^{-1}(\mathbf{p}_i, \mathbf{d}_i)), \qquad \mathbf{p}_{ij} \in \mathbb{R}^{H \times W \times 2} \qquad \mathbf{G}_{ij} = \mathbf{G}_j \circ \mathbf{G}_i^{-1}. \tag{3}$$

for each edge $(i, j) \in \mathcal{E}$ in the frame graph. Here $\Pi_c$ is the camera model mapping a set of 3D points onto the image and $\Pi_c^{-1}$ is the inverse projection function mapping inverse depth map $\mathbf{d}$ and coordinate grid $\mathbf{p}_i$ to a 3D point cloud (we provide formulas and Jacobians in the appendix). $\mathbf{p}_{ij}$ represents the coordinates of pixels $\mathbf{p}_i$ mapped into frame $j$ using the estimated pose and depth.

**Inputs** We use the correspondence field to index the correlation volumes. For each edge $(i, j) \in \mathcal{E}$ we use $\mathbf{p}_{ij}$ to perform lookup from the correlation volume $\mathbf{C}_{ij}$ to retrieve correlation features. Additionally, we use the correspondence field to derive optical flow induced by camera motion as the difference $\mathbf{p}_{ij} - \mathbf{p}_j$. Furthermore, the residual from the previous BA solution is concatenated with the flow field allowing the network to use feedback from the previous iteration.

The correlation features provide information about visual similarity in the neighbourhood of $\mathbf{p}_{ij}$ allowing the network to learn to align visually similar image regions. However, correspondence is sometimes ambiguous. The flow provides an complementary source of information allowing the network to exploit smoothness in the motion fields to gain robustness.

**Update** The correlation features and flow features are each mapped through two convolutional layers before being injected into the GRU. Additionally, we inject context features, as extracted by the context network, into the GRU through element-wise addition.

The ConvGRU is a local operation with a small receptive field. We extract global context by averaging the hidden state across the spatial dimensions of the image and use this feature vector as additional input to the GRU. Global context is important in SLAM because incorrect correspondences, caused by large moving objects for example, can degrade the accuracy of the system. It is important for the network to recognize and reject erroneous correspondence.

The GRU produces an updated hidden state $\mathbf{h}^{(k+1)}$. Instead of predicting updates to the depth or pose directly, we instead predict updates in the space of dense flow fields. We map the hidden state through two additional convolution layers to produce two outputs: (1) a revision flow field $\mathbf{r}_{ij} \in \mathbb{R}^{H \times W \times 2}$ and (2) associated confidence map $\mathbf{w}_{ij} \in \mathbb{R}_+^{H \times W \times 2}$. The revision $\mathbf{r}_{ij}$ is a correction term predicted by the network to correct errors in the dense correspondence field. We denote the corrected correspondence as $\mathbf{p}_{ij}^* = \mathbf{r}_{ij} + \mathbf{p}_{ij}$

We then pool the hidden state over all features which share the same source view $i$ and predict a pixel-wise damping factor $\lambda$. We use the softplus operator to ensure that the damping term is positive. Additionally, we use the pooled features to predict a 8x8 mask which can be used to upsample the inverse depth estimate.

**Dense Bundle Adjustment Layer (DBA)** The Dense Bundle Adjustment Layer (DBA) maps the set of flow revisions into a set of pose and pixelwise depth updates. We define the cost function over the entire frame graph

$$\mathbf{E}(\mathbf{G}', \mathbf{d}') = \sum_{(i,j) \in \mathcal{E}} \left\| \mathbf{p}_{ij}^* - \Pi_c(\mathbf{G}_{ij}' \circ \Pi_c^{-1}(\mathbf{p}_i, \mathbf{d}_i')) \right\|_{\Sigma_{ij}}^2 \qquad \Sigma_{ij} = \operatorname{diag} \mathbf{w}_{ij}. \qquad (4)$$

where $\|\cdot\|_\Sigma$ is the Mahalanobis distance which weights the error terms based on the confidence weights $\mathbf{w}_{ij}$. Eqn. 4 states that we want an updated pose $\mathbf{G}'$ and depth $\mathbf{d}'$ such that reprojected points match the revised correspondence $\mathbf{p}_{ij}^*$ as predicted by the update operator.

We use local parameterization to linearize Eqn. 4 and use the Gauss-Newton algorithm solve for updates $(\Delta \boldsymbol{\xi}, \Delta \mathbf{d})$. Since each term in Eqn. 4 only includes a single depth variable, the Hessian matrix has block diagonal structure. Separating pose and depth variables, the system can be solved efficiently using the Schur complement with the pixelwise damping factor $\lambda$ added to the depth block

$$\begin{bmatrix} \mathbf{B} & \mathbf{E} \\ \mathbf{E}^T & \mathbf{C} \end{bmatrix} \begin{bmatrix} \Delta \boldsymbol{\xi} \\ \Delta \mathbf{d} \end{bmatrix} = \begin{bmatrix} \mathbf{v} \\ \mathbf{w} \end{bmatrix} \qquad \begin{aligned} \Delta \boldsymbol{\xi} &= [\mathbf{B} - \mathbf{E}\mathbf{C}^{-1}\mathbf{E}^T]^{-1}(\mathbf{v} - \mathbf{E}\mathbf{C}^{-1}\mathbf{w}) \\ \Delta \mathbf{d} &= \mathbf{C}^{-1}(\mathbf{w} - \mathbf{E}^T \Delta \boldsymbol{\xi}) \end{aligned} \qquad (5)$$

where $\mathbf{C}$ is diagonal and can be cheaply inverted $\mathbf{C}^{-1} = 1/\mathbf{C}$. The DBA layer is implemented as part of the computation graph and backpropagation is performed through the layer during training.

### 3.3 Training

Our SLAM system is implemented in PyTorch and we use the LieTorch extension [49] to perform backprogration in the tangent space of all group elements.

**Removing gauge freedom** In the monocular setting, the network is only able to recover the trajectory of the camera up to a similarity transform. One solution is to define a loss which is invariant to similarity transforms. However, the gauge-freedom still exists during training which poorly impacts the conditioning of the linear system and the stability of the gradients. We solve this problem by fixing the first two poses to the ground-truth poses of each training sequence. Fixing the first pose removes the 6-dof gauge freedom. Fixing the second pose resolves the scale freedom.

**Constructing training video** Each training example consists of a 7-frame video sequence. In order to ensure stable training and good downstream performance, we want to sample videos which are not too easy nor too difficult.

The training set is composed of a collection of videos. For each video $i$ of length $N_i$, we precompute an $N_i \times N_i$ distance matrix storing the average optical flow magnitude between each pair of frames. However, not all frames are covisible; and frames pairs with less than 50% overlap are assigned a distance of infinity. During training, we dynamically generate videos by sampling paths in the distance matrix, such that the average flow between adjacent video frames is between 8px and 96px.

**Supervision** We supervise our network using a combination of *pose* loss and *flow* loss. The flow loss is applied to pairs of adjacent frames. We compute the optical flow induced by the predicted depth and poses and the flow induced by the ground truth depth and poses. The loss is taken to be the average l2 distance between the two flow fields.

Given a set of ground truth poses $\{\mathbf{T}\}_i^N$ and predicted poses $\{\mathbf{G}\}_i^N$, the pose loss is taken to be the distance between the ground truth and predicted poses, $\mathcal{L}_{pose} = \sum_i || \operatorname{Log}_{SE3}(\mathbf{T}_i^{-1} \cdot \mathbf{G}_i)||_2$. We apply the losses to the output of every iteration with exponentially increasing weight using $\gamma = 0.9$.

## 3.4 SLAM System

During inference, we compose the network into a full SLAM system. The SLAM system takes a video stream as input, and performs reconstruction and localization in real-time. Our system contains two threads which run asynchronously. The *frontend* thread takes in new frames, extracts features, selects keyframes, and performs local bundle adjustment. The *backend* thread simultaneously performs global bundle adjustment over the entire history of keyframes. We provide an overview of the system here, and provide more information in the appendix.

**Initialization** Initialization is simple with DROID-SLAM. We simply collect frames until we have a set of 12. As we accumulate frames, we only keep the previous frame if optical flow is greater than 16px (estimated by applying one update iteration). Once 12 frames have been accumulated, we initialize a frame graph by creating an edges between keyframes which are within 3 timesteps apart, then run 10 iterations of the update operator.

**Frontend** The frontend operates directly on the incoming video stream. It maintains a collection of keyframes and a frame graph storing edges between covisible kefyrames. Keyframe poses and depths are actively being optimized. Features are first extracted from the incoming frames. The new frame is then added to the frame graph adding edges with its 3 closest neighbors as measured by mean optical flow. The pose is initialized using a linear motion model. We then apply several iterations of the update operator to update keyframe poses and depths. We fix the first two poses to remove gauge freedom but treat all depths as free variables.

After the new frame is tracked, we select a keyframe for removal. We compute distance between pairs of frames by computing the average optical flow magnitude and remove redundant frames. If no frame is a good candidate for removal, we remove the oldest keyframe.

**Backend** The backend performs global bundle adjustment over the entire history of keyframes. During each iteration, we rebuild the frame graph using the flow between all pairs of keyframes, represented as an $N \times N$ distance matrix. We first add edges between temporally adjacent keyframes. We then sample new edges from the distance matrix in order of increasing flow. With each selected edge, we suppress neighboring edges within a distance of 2, where distance is defined as the Chebyshev distance between index pairs $||(i, j) - (k, l)||_\infty = \max(|i - k|, |j - l|)$.

We then apply the update operator to the entire frame graph, often consisting of thousands of frames and edges. Storing the full set of correlation volumes would quickly exceed video memory. Instead, we use the memory efficient implementation proposed in RAFT [48].

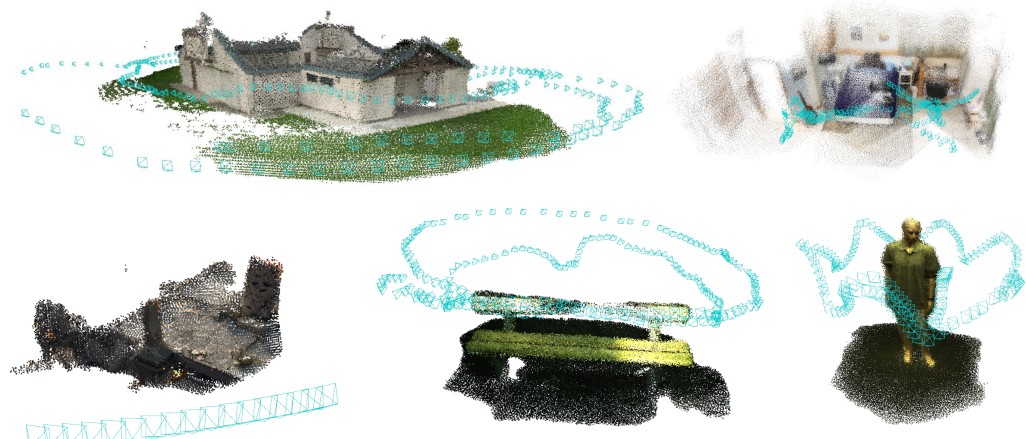

Figure 3: DROID-SLAM can generalize to new datasets. In order, we show results from Tanks & Temples [21], ScanNet [10], Sintel [3], and ETH-3D [41]; all using monocular video.

During training, we implement dense bundle adjustment in PyTorch to leverage the automatic differentiation engine. At inference time, we use a custom CUDA kernel which takes advantage of the block-sparse structure of the problem, then perform sparse Cholesky decomposition on the reduced camera block.

We only perform full bundle adjustment on keyframe images. In order to recover the poses of non-keyframes, we perform motion-only bundle adjustment by iteratively estimating flow between each keyframe and its neighboring non-keyframes. During testing, we evaluate on the full camera trajectory, not just keyframes.

**Stereo and RGB-D** Our system can be easily modified for stereo and RGB-D video. In the case of RGB-D, we still treat depth as a variable, since sensor depth can be noisy and have missing observations, and simply add a term to the optimization objective (Eqn. 4) which penalizes the squared distance between the measured and predicted depth. For stereo, we use the exact same system described above, with just double the frames, and fix the relative pose between the left and right frames in the DBA layer. Cross camera edges in the graph allow us to leverage stereo information.

## 4 Experiments

We experiment on a diverse set of datasets and sensor modalities. We compare to both deep learning and established classical SLAM algorithms and put specific emphasis on cross-dataset generalization. Following prior work, we evaluate the accuracy of the camera trajectory [31, 15, 41], primarily using Absolute Trajectory Error (ATE) [43]. While some datasets have ground truth point clouds [21], there is no standard protocol to compare 3D reconstructions directly given by SLAM systems because a SLAM systems can choose which 3D points to reconstruct. Evaluating dense 3D reconstruction is typically considered in the domain of Multiview Stereo [19] and outside the scope of this work.

Our network is trained entirely on monocular video from the synthetic TartanAir dataset [54]. We train our network for 250k steps with a batch size of 4, resolution $384 \times 512$, and 7 frame clips, and unroll 15 update iterations. Training takes 1 week on 4 RTX-3090 GPUs.

| Monocular | MH000 | MH001 | MH002 | MH003 | MH004 | MH005 | MH006 | MH007 | Avg |
|---|---|---|---|---|---|---|---|---|---|
| ORB-SLAM [31] | 1.30 | **0.04** | 2.37 | 2.45 | X | X | 21.47 | 2.73 | - |
| DeepV2D [47] | 6.15 | 2.12 | 4.54 | 3.89 | 2.71 | 11.55 | 5.53 | 3.76 | 5.03 |
| TartanVO [53] | 4.88 | 0.26 | 2.00 | 0.94 | 1.07 | 3.19 | 1.00 | 2.04 | 1.92 |
| Ours | **0.08** | 0.05 | **0.04** | **0.02** | **0.01** | **1.31** | **0.30** | **0.07** | **0.24** |

Table 1: Results on the TartanAir monocular benchmark.

**TartanAir [54] (Monocular & Stereo)** The TartanAir dataset is a challenging synthetic benchmark for evaluating SLAM algorithms and was used as part of the ECCV 2020 SLAM competition. We use the official test split [54], and provide ATE across all "Hard" sequences in Tab. 1.

Tab. 1 demonstrates both the robustness of our method (no catastrophic failures) and accuracy (very low drift). We retrain DeepV2D [47] on TartanAir as a baseline. On most sequences, we outperform existing methods by an order-of-magnitude and achieve 8x lower average error than TartanVO [53] and 20x lower than DeepV2D [47]. We also use the TartanAir dataset to compare with the top submissions to the ECCV 2020 SLAM competition in Tab. 2. The top two submissions use systems

|  | Mono. | Stereo |
|---|---|---|
| OV$^2$SLAM [17] | 0.510 | 0.182 |
| VOLDOR [28] + COLMAP [40] | 0.440 | 0.177 |
| SuperGlue [38] + SuperPoint [13] + COLMAP [40] | 0.340 | 0.119 |
| Ours | **0.129** | **0.047** |

Table 2: Results on the TartanAir test set, compared with the top 3 submission to the ECCV 2020 SLAM competition. The score is computed using normalized relative pose error for all possible sequences of length $\{5, 10, 15, ..., 40\}$ meters, see competition page for details.

built on top of COLMAP [40] and run 40x slower than real-time. Our method, on the other hand, runs 16x faster and achieves an error 62% lower on the monocular benchmark and 60% lower on the stereo benchmark.

**EuRoC [2] (Monocular & Stereo)** In the remaining experiments, we are interested in the ability of our network to generalize to new cameras and environments. The EuRoC dataset consists of video captured from sensor on-board a micro aerial vehicle (MAV) and is a widely used benchmark to evaluate SLAM systems. We use the EuRoC dataset to evaluate both monocular and stereo performance and report results on Tab. 3.

| | | MH01 | MH02 | MH03 | MH04 | MH05 | V101 | V102 | V103 | V201 | V202 | V203 | Avg |
|---|---|---|---|---|---|---|---|---|---|---|---|---|---|
| Deep/Hyb. | DeepFactors [9] | 1.587 | 1.479 | 3.139 | 5.331 | 4.002 | 1.520 | 0.679 | 0.900 | 0.876 | 1.905 | 1.021 | 2.040 |
| | DeepV2D [47][†] | 0.739 | 1.144 | 0.752 | 1.492 | 1.567 | 0.981 | 0.801 | 1.570 | 0.290 | 2.202 | 2.743 | 1.298 |
| | DeepV2D (Tartan Air)[†] | 1.614 | 1.492 | 1.635 | 1.775 | 1.013 | 0.717 | 0.695 | 1.483 | 0.839 | 1.052 | 0.591 | 1.173 |
| | TartanVO[1] [53][†] | 0.639 | 0.325 | 0.550 | 1.153 | 1.021 | 0.447 | 0.389 | 0.622 | 0.433 | 0.749 | 1.152 | 0.680 |
| | D3VO + DSO [57][†] | - | - | 0.08 | - | 0.09 | - | - | 0.11 | - | 0.05 | 0.19 | - |
| Classical | ORB-SLAM [31] | 0.071 | 0.067 | 0.071 | 0.082 | 0.060 | **0.015** | 0.020 | X | 0.021 | 0.018 | X | - |
| | DSO [15][†] | 0.046 | 0.046 | 0.172 | 3.810 | 0.110 | 0.089 | 0.107 | 0.903 | 0.044 | 0.132 | 1.152 | 0.601 |
| | SVO [18][†] | 0.100 | 0.120 | 0.410 | 0.430 | 0.300 | 0.070 | 0.210 | X | 0.110 | 0.110 | 1.080 | - |
| | DSM [60] | 0.039 | 0.036 | 0.055 | 0.057 | 0.067 | 0.095 | 0.059 | 0.076 | 0.056 | 0.057 | 0.784 | 0.126 |
| | ORB-SLAM3 [5] | 0.016 | 0.027 | 0.028 | 0.138 | 0.072 | 0.033 | 0.015 | 0.033 | 0.023 | 0.029 | X | - |
| | Ours (odometry only)[†] | 0.163 | 0.121 | 0.242 | 0.399 | 0.270 | 0.103 | 0.165 | 0.158 | 0.102 | 0.115 | 0.204 | 0.186 |
| | Ours | **0.013** | **0.014** | **0.022** | **0.043** | **0.043** | 0.037 | **0.012** | **0.020** | **0.017** | **0.013** | **0.014** | **0.022** |

Table 3: Monocular SLAM on the EuRoC datasets, ATE[m]. [†] denotes visual odometry methods.

In the monocular setting, we achieve an average ATE of 2.2cm, reducing error by 82% among methods with zero failures, and by 43% over ORB-SLAM3 when only comparing sequences where ORB-SLAM3 is successful.

We compare to several deep learning approaches. We compare to DeepV2D trained on the TartanAir dataset and the publicly available version trained on NYUv2 [33] and ScanNet[10]. DeepFactors [9] was trained on ScanNet. We find that recent deep learning approaches [9, 47, 53] perform poorly on the EuRoC dataset compared to classical SLAM systems. This is due to poor generalization and dataset biases which lead to large amounts of drift; our method does not suffer from these issues. D3VO [57] is able to achieve both good robustness and accuracy by combining a neural network frontend with DSO as a backend, using 6 of the 11 sequences for evaluation and performing unsupervised training on the remaining ones, which contain the same scenes used for evaluation.

**TUM-RGBD [43]** The RGBD dataset consists of indoor scenes captured with handheld camera. This is a notoriously difficult dataset for monocular methods due to rolling shutter artifacts, motion blur, and heavy rotation. We benchmark prior work on the entirety of the freiburg1 set in Tab. 4.

Classical SLAM algorithms such as ORB-SLAM tend to fail on most of the sequences. While deep learning methods are more robust, they obtain low accuracy on most of the evaluated sequences. Our

| | 360 | desk | desk2 | floor | plant | room | rpy | teddy | xyz | avg |
|---|---|---|---|---|---|---|---|---|---|---|
| ORB-SLAM2 [32] | X | 0.071 | X | 0.023 | X | X | X | X | 0.010 | - |
| ORB-SLAM3 [5] | X | **0.017** | 0.210 | X | 0.034 | X | X | X | **0.009** | - |
| DeepTAM[1] [59] | 0.111 | 0.053 | 0.103 | 0.206 | 0.064 | 0.239 | 0.093 | 0.144 | 0.036 | 0.116 |
| TartanVO[2] [53] | 0.178 | 0.125 | 0.122 | 0.349 | 0.297 | 0.333 | 0.049 | 0.339 | 0.062 | 0.206 |
| DeepV2D [47] | 0.243 | 0.166 | 0.379 | 1.653 | 0.203 | 0.246 | 0.105 | 0.316 | 0.064 | 0.375 |
| DeepV2D (TartanAir) | 0.182 | 0.652 | 0.633 | 0.579 | 0.582 | 0.776 | 0.053 | 0.602 | 0.150 | 0.468 |
| DeepFactors [9] | 0.159 | 0.170 | 0.253 | 0.169 | 0.305 | 0.364 | 0.043 | 0.601 | 0.035 | 0.233 |
| Ours | **0.111** | 0.018 | **0.042** | **0.021** | **0.016** | **0.049** | **0.026** | **0.048** | 0.012 | **0.038** |

Table 4: ATE on the TUM-RGBD benchmark. All methods are provided mono. video, [1]except DeepTAM which uses RGB-D and [2]TartanVO which uses ground truth to scale relative pose.

method is both robust and accurate. It successfully tracks all 9 sequences while achieving 83% lower ATE than DeepFactors [9] and which succeeds on all videos and 90% lower ATE than DeepV2D [47].

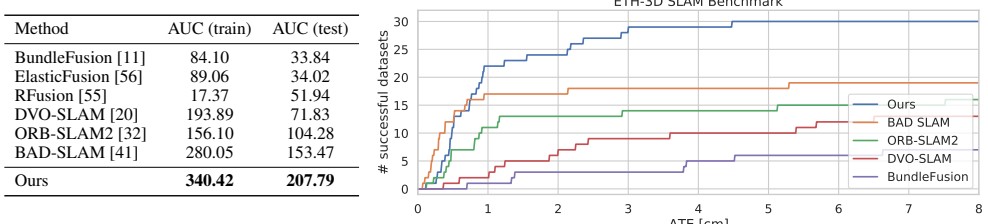

| Method | AUC (train) | AUC (test) |
|---|---|---|
| BundleFusion [11] | 84.10 | 33.84 |
| ElasticFusion [56] | 89.06 | 34.02 |
| RFusion [55] | 17.37 | 51.94 |
| DVO-SLAM [20] | 193.89 | 71.83 |
| ORB-SLAM2 [32] | 156.10 | 104.28 |
| BAD-SLAM [41] | 280.05 | 153.47 |
| Ours | **340.42** | **207.79** |

Figure 4: Generalization results on the RGB-D ETH3D-SLAM benchmark. (Left) Our method, which is trained only on the synthetic TartanAir dataset, ranks 1st on both the train and test splits. (Right) Plot of the number successful trajectories as a function of ATE. Our method successfully tracks 30/32 of the datasets where image data is available.

**ETH3D-SLAM [41] (RGB-D)** Finally, we evaluate the RGB-D performance on the ETH3D-SLAM benchmark. In this setup, the network is also provided measurements from an RGB-D camera. We take our network trained on TartanAir and add an addition term in the optimization objective penalizing the distance between the predicted inv. depth and inv. depth measured by the sensor. Without any finetuning, our method ranks 1st on both the train and test splits. Several of the datasets are "dark" meaning no image data is available; on these datasets we do not submit any predictions. On the test set, we successfully track 30/32 RGB-D, improving over the next best of 19/32.

**Timing and Memory** Our system can run in real-time with 2 3090 GPUs. Tracking and local BA is run on the first GPU, while global BA and loop closure is run on the second. On EuRoC, we average 20fps (camera hz) by downsampling to $320 \times 512$ resolution and skipping every other frame. Results in Tab. 3 were obtained in this setting. On TUM-RGBD, we average 30fps by downsampling to $240 \times 320$ and skipping every other frame, again the reported results where obtained in this setting. On TartanAir, due to much faster camera motion, we are unable to run in real-time, averaging 8fps. However, this is still a 16x speedup over the top 2 submissions to the TartanAir SLAM challenge, which rely on COLMAP [40].

The SLAM frontend can be run on GPUs with 8GB of memory. The backend, which requires storing feature maps from the full set of images, is more memory intensive. All results on TUM-RGBD can be produced on a single 1080Ti graphics card. Results on EuRoC, TartanAir and ETH-3D (where video can be up to 5000 frames) requires a GPU with 24GB memory. While memory and resource requirements are currently the biggest limitation of our system, we believe these can be drastically reduced by culling redundant computation and more efficient representations.

## 5 Conclusion

We introduce DROID-SLAM, an end-to-end neural architecture for visual SLAM. DROID-SLAM is accurate, robust, and versatile and can be used on monocular, stereo, and RGB-D video. It outperforms prior work by large margins on challenging benchmarks.

**Acknowledgements** This work is partially supported by the National Science Foundation under Award IIS-1942981.

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
