# DROID-SLAM: Supplementary Material

## A  Additional Results

|  | MH01 | MH02 | MH03 | MH04 | MH05 | V101 | V102 | V103 | V201 | V202 | V203 | Avg |
|---|---|---|---|---|---|---|---|---|---|---|---|---|
| D3VO + DSO [6] | - | - | 0.08 | - | 0.09 | - | - | 0.11 | - | 0.05 | - | - |
| ORB-SLAM2 [4] | 0.035 | 0.018 | 0.028 | 0.119 | 0.060 | **0.035** | 0.020 | 0.048 | 0.037 | 0.035 | - | - |
| VINS-Fusion [5] | 0.540 | 0.460 | 0.330 | 0.780 | 0.500 | 0.550 | 0.230 | - | 0.230 | 0.200 | - | - |
| SVO [3] | 0.040 | 0.070 | 0.270 | 0.170 | 0.120 | 0.040 | 0.040 | 0.070 | 0.050 | 0.090 | 0.790 | 0.159 |
| ORB-SLAM3 [2] | 0.029 | 0.019 | **0.024** | 0.085 | 0.052 | **0.035** | 0.025 | 0.061 | 0.041 | 0.028 | 0.521 | 0.084 |
| Ours | **0.015** | **0.013** | 0.035 | **0.048** | **0.040** | 0.037 | **0.011** | **0.020** | **0.018** | **0.015** | **0.017** | **0.024** |

Table 1: Stereo SLAM on the EuRoC datasets, ATE[m].

We provide stereo results on the EuRoC dataset[1] in Tab. 1 using our network trained on synthetic, monocular video. In the stereo setting, it is possible to recover the trajectory of the camera up to scale. Compared to ORB-SLAM3[2] we reduce the average ATE by 71%.

## B  Ablations

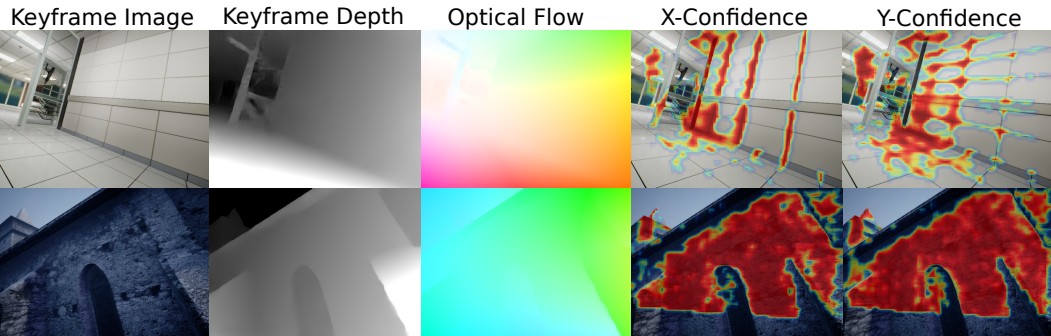

Figure 1: Visualizations of keyframe image, depth, flow and confidence estimates.

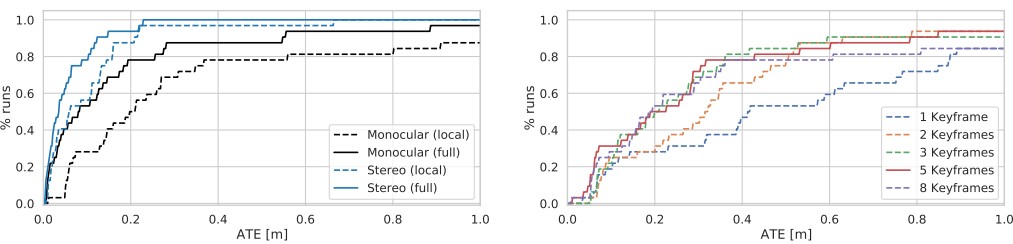

Figure 2: (Left) we show the performance of the system with different inputs (monocular vs. stereo) and whether global optimization is performed in addition to local BA (local vs. full). (Right) Tracking accuracy as a function of the number of keyframes. We use 5 keyframes (bold) in our experiments.

**Ablations** We ablate various design choices regarding our SLAM system and network architecture. Ablations are performed on our validation split of the TartanAir dataset. In Fig. 1 we show visualizations on the validation set of keyframe depth estimates alongside optical flow and associated confidence weights.

In Fig.2 (left) we show how the system benefits from both stereo video and global optimization. Although our network is only trained on monocular video, it can readily leverage stereo frames if available. In Fig. 2 (right) we show how the number of keyframe affects odometery performance. In Fig. 3 we ablate components of the network architecture. Fig. 3 (left) shows the impact of using global context in the GRU through spatial pooling while 3 (right) demonstrates the importance of

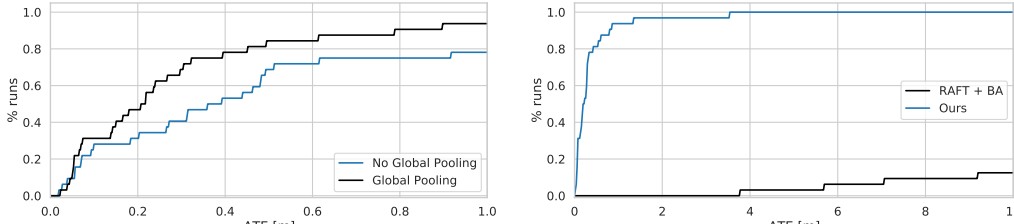

Figure 3: (Left) Impact of global context in the update operator. (Right) Impact of using the bundle adjustment layer during training vs training directly on optical flow, then applying BA at test time.

training with DBA as opposed to training on flow and applying BA at inference. We find that the SLAM system is unstable and prone to failure if the DBA is not used during training.

## C   Camera Model and Jacobians

We represent 3D points using homogeneous coordinates $\mathbf{X} = (X, Y, Z, W)^T$. An image point $\mathbf{p}$ with inverse depth $d$ is re-projected from frame $i$ into frame $j$ according to the warping function

$$\mathbf{p}' = \Pi_c(\mathbf{G}_{ij} \cdot \Pi^{-1}(\mathbf{p}, \mathbf{d})) \qquad \mathbf{G}_{ij} = \mathbf{G}_j \circ \mathbf{G}_i^{-1} \tag{1}$$

where $\Pi_c$ is the pinhole projection function, and $\Pi_c^{-1}$ is the inverse projection

$$\Pi_c(\mathbf{X}) = \begin{pmatrix} f_x \frac{X}{Z} + c_y \\ f_y \frac{Y}{Z} + c_y \end{pmatrix} \qquad \Pi_c^{-1}(\mathbf{p}, d) = \begin{pmatrix} \frac{p_x - c_x}{f_x} \\ \frac{p_y - c_y}{f_y} \\ 1 \\ d \end{pmatrix}. \tag{2}$$

given camera intrinsic parameters $c = (f_x, f_y, c_x, c_y)$.

For optimization, we need the Jacobians with respect to $\mathbf{G}_i$, $\mathbf{G}_j$, and $d$. We use the local parameterization $e^{\xi_i} \mathbf{G}_i$ and $e^{\xi_j} \mathbf{G}_j$ and treat $d$ as a vector in $\mathbb{R}^1$. The Jacobians of the projection and inverse projection functions are given as

$$\frac{\partial \Pi_c(\mathbf{X})}{\partial \mathbf{X}} = \begin{pmatrix} f_x \frac{1}{Z} & 0 & -f_x \frac{X}{Z^2} & 0 \\ 0 & f_y \frac{1}{Z} & -f_y \frac{Y}{Z^2} & 0 \end{pmatrix} \qquad \frac{\partial \Pi_c^{-1}(\mathbf{p}, d)}{\partial d} = \begin{pmatrix} 0 \\ 0 \\ 0 \\ 1 \end{pmatrix}. \tag{3}$$

Using the local parameterization, we compute the Jacobian of the 3D point transformation

$$\mathbf{X}' = \operatorname{Exp}(\xi_j) \cdot \mathbf{G}_j \cdot (\operatorname{Exp}(\xi_i) \cdot \mathbf{G}_i)^{-1} \cdot \mathbf{X} = \operatorname{Exp}(\xi_j) \cdot \mathbf{G}_j \cdot \mathbf{G}_i^{-1} \cdot \operatorname{Exp}(-\xi_i) \cdot \mathbf{X} \tag{4}$$

using the adjoint operator to move the $\xi_i$ term to the front of the expression

$$\mathbf{X}' = \operatorname{Exp}(\xi_j) \cdot \operatorname{Exp}(-\operatorname{Adj}_{\mathbf{G}_j \mathbf{G}_i^{-1}} \xi_i) \cdot \mathbf{G}_j \cdot \mathbf{G}_i^{-1} \cdot \mathbf{X} \tag{5}$$

allowing us to compute the Jacobians using the generators

$$\frac{\partial \mathbf{X}'}{\partial \xi_j} = \begin{pmatrix} W' & 0 & 0 & 0 & Z' & -Y' \\ 0 & W' & 0 & -Z' & 0 & X' \\ 0 & 0 & W' & Y' & -X' & 0 \\ 0 & 0 & 0 & 0 & 0 & 0 \end{pmatrix} \tag{6}$$

$$\frac{\partial \mathbf{X}'}{\partial \xi_i} = - \begin{pmatrix} W' & 0 & 0 & 0 & Z' & -Y' \\ 0 & W' & 0 & -Z' & 0 & X' \\ 0 & 0 & W' & Y' & -X' & 0 \\ 0 & 0 & 0 & 0 & 0 & 0 \end{pmatrix} \cdot \operatorname{Adj}_{\mathbf{G}_j \mathbf{G}_i^{-1}} \tag{7}$$

Using the chain rule, we can compute the full Jacobians with respect to the variables

$$\frac{\partial \mathbf{p}'}{\partial \xi_j} = \frac{\partial \Pi_c(\mathbf{X}')}{\partial \mathbf{X}'} \frac{\partial \mathbf{X}'}{\partial \xi_j}, \qquad \frac{\partial \mathbf{p}'}{\partial \xi_i} = \frac{\partial \Pi_c(\mathbf{X}')}{\partial \mathbf{X}'} \frac{\partial \mathbf{X}'}{\partial \xi_i} \qquad (8)$$

$$\frac{\partial \mathbf{p}'}{\partial d} = \frac{\partial \Pi_c(\mathbf{X}')}{\partial \mathbf{X}'} \frac{\partial \mathbf{X}'}{\partial \mathbf{X}} \frac{\partial \Pi^{-1}(\mathbf{p}, d)}{\partial d} = \frac{\partial \Pi_c(\mathbf{X}')}{\partial \mathbf{X}'} = \frac{\partial \Pi_c(\mathbf{X}')}{\partial \mathbf{X}'} \begin{pmatrix} t_x \\ t_y \\ t_z \\ 1 \end{pmatrix} \qquad (9)$$

where $(t_x, t_y, t_z)$ is the translation vector of $\mathbf{G}_j \circ \mathbf{G}_i^{-1}$.

## D   Network Architecture

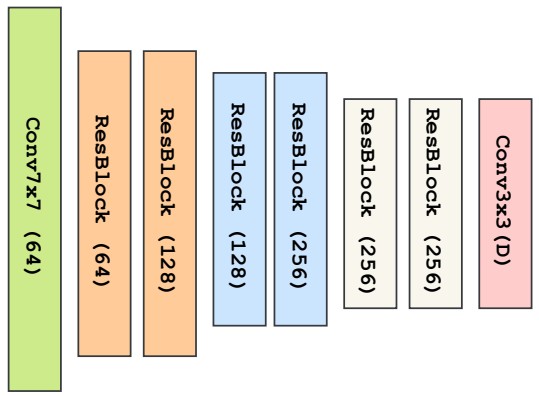

Figure 4: Architecture of the feature and context encoders. Both extract features at 1/8 the input image resolution using a set of 6 basic residual blocks. Instance normalization is used in the feature encoder; no normalization is used in the context encoder. The feature encoder outputs features with dimension D=128 which the context encoder outputs features with dimension D=256.

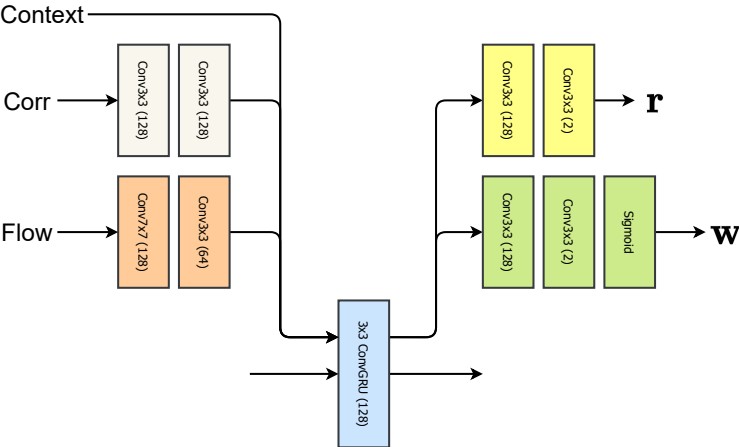

Figure 5: Architecture of the update operator. During each iteration, context, correlation, and flow features get injected into the GRU. The revision (r) and confidence weights (w) are predicted from the updated hidden state.