# OpenReview forum: "DROID-SLAM: Deep Visual SLAM for Monocular, Stereo, and RGB-D Cameras"
_NeurIPS.cc/2021/Conference — NeurIPS 2021 Oral_

### Official Review · Reviewer_GZge · 2021-06-28

**Rating:** 8
**Confidence:** 5

**Summary:**

This paper presents R-SLAM, a learning-based SLAM method that shows state-of-the-art camera pose estimation performance on several benchmarks. In particular, R-SLAM updates the camera poses and dense frame depth in an iterative manner by using a recurrent update operator built upon RAFT and differentiable bundle adjustment. To make a complete SLAM system, R-SLAM imposes a covisibility graph and conducts both local and global optimization. The experiments show that R-SLAM outperforms the previous classical and learning-based approaches by a large margin, even on the datasets on which it is not trained.

**Limitations And Societal Impact:**

The authors have addressed memory consumption as one of the limitations of the method.

**Main Review:**

## Strength
1. The proposed SLAM method, R-SLAM, has shown the state-of-the-art camera pose estimation results on several established benchmarks, including TantanAir, EuRoC, TUM RGB-D, and ETH3D. Also, R-SLAM shows in general quite large improvements over the prior works, e.g., by ~60% on TantanAir, ~40% on EuRoC, ~80% on TUM RGB-D, and ~35% on ETH3D.
2. The state-of-the-art results demonstrated are obtained with the model trained on TantanAir only, thus showing very strong generalization capability which is often the major limitation of learning-based SLAM approaches.
3. R-SLAM takes advantage of both classical optimization and deep learning -- it can be trained end-to-end while keeping the bundle adjustment (BA) with Gauss-Newton in the loop. Besides, R-SLAM conducts the joint optimization of pose and depth which has been shown to be one of the keys to accurate estimations.
4. Besides the high accuracy, R-SLAM also shows improved robustness with a high success rate on several datasets.

## Weaknesses
1. In general, the novelty of the paper is not substantial. It builds upon the established systems and methodologies -- a hybrid SLAM design that combines both the classical and learning worlds, differentiable BA / Gauss-Newton layer, covisibility graph, and the recurrent update operator. Overall it feels like a system as DeepV2D + RAFT, and no fundamental theoretical or architectural innovations are proposed. Above being said, I believe combining the previous works and making the right choices that lead to notable improvements is still quite valuable and inspiring to the community.
2. R-SLAM is trained on TatanAir while most of the compared methods (DeepFactors, DeepV2D, D3VO, DeepTAM) are not. Although some of the aforementioned methods are trained with the train/test split on the same data while R-SLAM shows mostly the cross-dataset generalization, it still poses a certain level of unfairness on the comparisons since, in general, the TantanAir dataset contains more images and more diversities. As a suggestion, it would be better to show a method trained and evaluated using exactly the same dataset as R-SLAM and report the comparison, for which DeepV2D could be a good baseline.
3. It is unclear how the non-keyframe poses are estimated if they are estimated at all. Since it is a SLAM system instead of an SfM system, R-SLAM should be able to deliver both keyframe and non-keyframe poses for real-world applications. However, from the paper one cannot see how the non-keyframe poses are handled. In fact, this infers other issues on the evaluation -- R-SLAM is evaluated with the keyframe pose without non-keyframe poses, but is it also the case for the other compared methods?; are the poses used in the evaluations the outputs of the global BA? If so, how good are the poses from local bundle adjustment since some compared methods are Visual Odometry systems without any global BA, e.g, DeepV2D, D3VO, DSO, etc?
4. The details of the full dense bundle adjustment are missing. R-SLAM performs joint dense BA on pose and depth which is quite expensive, especially when doing the global BA which optimizes tens or even hundreds of frame depth and poses. Is there a customized GPU-based optimizer used?
5. It would be better if more qualitative results can be shown, e.g., camera trajectories comparison among the methods and the estimated dense depth maps.
6. The writing of the paper can still be improved. There is a number of typos in the current manuscript. I list some of them here but there should be more:
- L10: "focused" -> "focuses"
- L13: "approach" -> "approached"
- L15: "are have" -> "have"
- L20: "But despite" -> "Despite" -
- L57: "solves computes" -> "computes"
- L81: "is that features" -> remove
- L115: "operators" -> "operates"
- Eq. (5): $\triangle w$ -> $\triangle d$?
7. The name "R-SLAM" already exists in [1] and [2]. In order to avoid confusion in the future, I would suggest the authors proposing a new name of the system.

## Overall Comment
I recommend accepting the paper due to the impressive results and the reasonable design choices made by the authors. But I still would like to see the authors' responses to the points listed in the Weaknesses section above.

----------
References:
1. Balcılar, Muhammet, et al. "R-SLAM: Resilient localization and mapping in challenging environments." Robotics and Autonomous Systems 87 (2017): 66-80.
2. Mei, Christopher, et al. “RSLAM: A system for large-scale mapping in constant-time using stereo. International journal of computer vision 94 (2), 198-214, 2011

**Time Spent Reviewing:**

4

---

> ### Author Response · Authors · 2021-08-10
> **Thank you for your review. We address the individual points below.**
>
> **R-SLAM is trained on TatanAir while most of the compared methods (DeepFactors, DeepV2D, D3VO, DeepTAM) are not.** This is a good point and we will make it more clear in the paper what training data is used for each method. Of these methods (DeepFactors, DeepV2D, D3VO, DeepTAM) only DeepV2D has publicly available training code. Following your suggestion, we were able to retrain DeepV2D on TartanAir and report the updated results below: these will be added to our paper. We see that DeepV2D performs slightly better when trained on TartanAir but still significantly underperforms ours.
>
> | TartanAir | MH000 | MH001 | MH002 | MH003 | MH004 | MH005 | MH006 | MH007 | Avg  |
> |-----------|-------|-------|-------|-------|-------|-------|-------|-------|------|
> | DeepV2D   | 6.15  | 2.12  | 4.54  | 3.89  | 2.71  | 11.55 | 5.53  | 3.76  | 5.03 |
> | Ours      | 0.08  | 0.05  | 0.04  | 0.02  | 0.01  | 1.31  | 0.30  | 0.07  | 0.24 |
>
> | EuRoC                 | MH01  | MH02  | MH03  | MH04  | MH05  | V101  | V102  | V103  | V201  | V202  | V203  | Avg   |
> |-----------------------|-------|-------|-------|-------|-------|-------|-------|-------|-------|-------|-------|-------|
> | DeepV2D (NYU+Scannet) | 0.739 | 1.144 | 0.752 | 1.492 | 1.567 | 0.981 | 0.801 | 1.570 | 0.290 | 2.202 | 2.743 | 1.298 |
> | DeepV2D (TartanAir)   | 1.614 | 1.492 | 1.635 | 1.775 | 1.013 | 0.717 | 0.695 | 1.483 | 0.839 | 1.052 | 0.591 | 1.173 |
> | Ours                  | 0.014 | 0.013 | 0.022 | 0.049 | 0.043 | 0.036 | 0.011 | 0.017 | 0.019 | 0.011 | 0.013 | 0.023 |
>
> | TUM-RGBD              | 360   | desk  | desk2 | floor | plant | room  | rpy   | teddy | xyz   | Avg   |
> |-----------------------|-------|-------|-------|-------|-------|-------|-------|-------|-------|-------|
> | DeepV2D (NYU+Scannet) | 0.243 | 0.166 | 0.379 | 1.653 | 0.203 | 0.246 | 0.105 | 0.316 | 0.064 | 0.375 |
> | DeepV2D (TartanAir)   | 0.182 | 0.652 | 0.633 | 0.579 | 0.582 | 0.776 | 0.053 | 0.602 | 0.150 | 0.468 |
> | Ours                  | 0.111 | 0.018 | 0.042 | 0.021 | 0.016 | 0.049 | 0.026 | 0.048 | 0.012 | 0.038 |
>
>
> **Are the poses used in the evaluations the outputs of the global BA? If so, how good are the poses from local bundle adjustment since some compared methods are Visual Odometry systems without any global BA, e.g, DeepV2D, D3VO, DSO, etc?**
>
> We perform evaluation using our full system, including global bundle adjustment. The frontend is designed to be robust and not necessarily accurate. Its main purpose is to track incoming frames and provide a good initialization for global BA, which is ultimately necessary to correct for drift.
>
> In the paper, we do perform ablations where only the frontend is used (Fig. 5 left). Below, we show additional results on the EuRoC dataset where only the frontend is used, with no loop closure or global BA. Our frontend significantly outperforms DSO, SVO, and DeepV2D  in terms of average error and is significantly more robust than DSO.
>
> D3VO performs better than our frontend. However, the comparison with D3VO is somewhat tricky. D3VO requires unsupervised pretraining on videos of the same scene. On EuRoc, it only evaluates on some sequences while using others for pretraining, such that each test scene is seen during pretraining. The pretraining can be understood as a form of SfM with photometric loss, with the reconstruction results “stored” in a single-image depth network, which can help correct scale drift during test time for the same scene. This pretraining can enforce global consistency of depth because the optimization objective includes all pairs of frames of the pretraining video. Thus although D3VO does not perform any global BA on the test video, arguably it has already performed a form of global 3D reconstruction of the same scene in advance through pretraining. Since D3VO does not have code available, we were not able to test the performance when trained on external data like ours.
>
> | EuRoC               | MH01  | MH02  | MH03  | MH04  | MH05  | V101  | V102  | V103  | V201  | V202  | V203  | Avg   |
> |---------------------|-------|-------|-------|-------|-------|-------|-------|-------|-------|-------|-------|-------|
> | DeepV2D (TartanAir) | 1.614 | 1.492 | 1.635 | 1.775 | 1.013 | 0.717 | 0.695 | 1.483 | 0.839 | 1.052 | 0.591 | 1.173 |
> | DSO[15]             | 0.046 | 0.046 | 0.172 | 3.810 | 0.110 | 0.087 | 0.107 | 0.903 | 0.044 | 0.132 | 1.152 | 0.601 |
> | SVO[18]             | 0.100 | 0.120 | 0.410 | 0.430 | 0.300 | 0.070 | 0.210 | x     | 0.110 | 0.110 | 1.080 | -     |
> | D3VO[55]            | -     | -     | 0.08  | -     | 0.09  | -     | -     | 0.11  | -     | 0.05  | 0.19  | -     |
> | Ours (VO)           | 0.163 | 0.121 | 0.242 | 0.399 | 0.270 | 0.103 | 0.165 | 0.158 | 0.102 | 0.115 | 0.204 | 0.186 |
> | Ours (Full)         | 0.014 | 0.013 | 0.022 | 0.049 | 0.043 | 0.036 | 0.011 | 0.017 | 0.019 | 0.011 | 0.013 | 0.023 |
>
> **It is unclear how the non-keyframe poses are estimated if they are estimated at all.** Thank you for pointing this out. This is an important detail that was accidentally left out from our submission and will be added. Our system does output the poses for every frame, not just keyframes, and every frame is used for evaluation following standard protocols. Non-keyframe poses are estimated using motion-only bundle adjustment by iteratively estimating flow between each keyframe and its neighboring non-keyframes.
>
> **The details of the full dense bundle adjustment are missing. Is there a customized GPU-based optimizer used?** Yes, we write a custom GPU kernel to efficiently perform the Schur complement. We then perform sparse Cholesky decomposition on the reduced camera block using the CHOLMOD library. Block-sparse structure is maintained at every step in the process.
>
> We put a fair amount of effort into optimizing our BA implementation. While the number of variables is large, Bundle Adjustment can be parallelized on the GPU. Each iteration for 7 frames and 30 dense flow fields takes 1.2 ms. For 200 keyframes frames / 1000 dense flow fields, it takes 12 ms.
>
> **It would be better if more qualitative results can be shown, e.g., camera trajectories comparison among the methods and the estimated dense depth maps.** Thank you for this suggestion. We will add visualization of the depth maps, flow fields, and confidence weight. Additionally, we will show camera pose trajectories compared with other methods.
>
> **The name "R-SLAM" already exists.** Thank you for pointing this out. We will change the name of our method in the final version of our paper.

---

### Official Review · Reviewer_YJaQ · 2021-07-16

**Rating:** 9
**Confidence:** 3

**Summary:**

This paper does dense Structure from Motion using a data-driven dense Bundle Adjustment layer, that is optimized using flow fields. Experiments conducted show that this method outperforms the state of the art and is generalizable to new datasets (which is rare for a learning based method).

**Ethical Concerns:**

None.

**Limitations And Societal Impact:**

Some minor grammatical errors that need correction:
126: they
143: show
176: 'convoluation'
227: an
231: 'activly begin'

Around line 123, it is mentioned that the frame graph is built using frame co-visibility, but it is not immediately clear as to how this co-visibility is calculated. It is mentioned later in the paper (207) that this is done by thresholding optic flow. This also needs to be mentioned here (around line 123). It is also not very clear how optic flow is used while training - when presumably the network's own optic flow capabilities will be non-existent (at the beginning of training). Is a pre-trained optic flow network being used for this?

Supervision (209): mentions that a combination of pose loss and flow loss is used and that ground truth poses and depth is used. Does this mean that this network cannot be trained using monocular sequences? Or would you use self-supervised mono-depth techniques to get the depth?



**Main Review:**

This paper builds on the work done in the RAFT [46] paper, that generates dense optic-flow-fields using an iterative optimization procedure. A 4D dense, multi-scale correlation volume (that is a representation of regions of similarity) is built on pairs of images and the flow-field (and its associated uncertainty) between the pair of images is built up iteratively from this, using a recurrent (GRU) unit. This is like a data-driven, learnt optimization process, that iteratively approaches the correct optic flow given lots of video sequences. However, unlike the RAFT paper, this is not learnt using ground truth flow. In this work, because the structure and motion in the scene needs to be estimated, this is trained using ground truth depth and poses [this is not entirely clear and should be explained better that the ground truth depth and pose are required during training - if I understand it right]. The iterative updates to the optic flow from the GRU unit is in turn used by a Dense Bundle Adjustment (DBA) layer to iteratively arrive at a dense depth map and pose estimate for a pair of images. DBA uses the cost function over a sequence of frames, with flow error (between what is predicted from the GRU and what is obtained from the current estimates of depth and pose) that is optimized using a Gauss-Newton algorithm. It is an important distinction from prior work in this area like BA-net, that used photometric error for optimization and this work uses the flow error. This presumably leads to better robustness to lighting changes and camera blur, etc. The entire network is trained end-to-end using a combination of pose loss and flow loss.

Experiments conducted indicate the ability of the system to surpass SoTa. A network trained on the Tartan Air dataset generalizes to the EuRoC and TUM-RGBD datasets, and improves on the ATE on those datasets by as much as 83%.
This is also achieved in real-time or near-real-time run-time on desktop hardware.

This paper is well written and is an impressive result and a significant contribution to the field.

**Time Spent Reviewing:**

6

---

> ### Author Response · Authors · 2021-08-10
> **Thank you for your review. We address the individual points below.**
>
> **The frame graph is built using frame co-visibility, but it is not immediately clear as to how this co-visibility is calculated.** Thank you for this suggestion, we will describe the details of how the frame graph is constructed (as described in Sec. 3.4) upfront instead of later.
>
> **It is also not very clear how optic flow is used while training - when presumably the network's own optic flow capabilities will be non-existent.** During training, optical flow is computed as a function of the ground truth depth and pose which is used to construct the training examples. We will make this detail more clear in the paper. We do not use a pretrained optical flow network, our entire approach is trained from scratch.
>
> **Supervision: Does this mean that this network cannot be trained using monocular sequences?** Our network cannot be trained using monocular sequences alone, because it requires ground truth camera pose and depth for training. We did not explore unsupervised training in this work. However, our network is entirely trained on synthetic data, where perfect depth and pose can be easily obtained. We found training on synthetic data to be sufficient for good generalization on real data.

---

### Official Review · Reviewer_2zGu · 2021-07-18

**Rating:** 7
**Confidence:** 5

**Summary:**

This paper learns to estimate the camera poses and depth by interlacing the residual flow field estimation and the geometric bundle adjustment iteratively. At each iteration, a residual flow field is estimated pair-wisely between two frames, and then a bundle adjustment step is applied on all residual flow fields anchored at a keyframe to update the current camera poses and depth. The experiments demonstrate the superiority of the proposed method over existing learning based methods and conventional methods on various datasets.


**Limitations And Societal Impact:**

The computation limitation has been discussed in line-322. However, the limitation about outlier handling is not mentioned.
Conventionally, we can do outlier rejection after correspondences have been estimated and then recover camera poses and depth from bundle adjustment. However, in this paper, the flow estimation and bundle adjustment are interlaced. At each iteration, some outlier points such as dynamic objects will always be projected incorrectly and further affect residual flow estimation and bundle adjustment. Unless the weight prediction has been demonstrated to be sufficiently powerful, otherwise it will also be a limitation for this method.

**Main Review:**

Strength:
1. Interlacing between 2D residual flow and 3D camera poses and depth is a nice design to tackle the SLAM problem.
2. The experiments are comprehensive and adequate. The results on various datasets demonstrate the generalization ability of the residual flow estimation.

Weakness:

In general, there is no critical weakness except for some issues that need to be further clarified:
1. The paper does not mention the training data source. It describes how the data is constructed in sec.3.3, but what are the datasets that are used to produce the training data?  Are they from the same dataset as evaluation or they are completely different such as a synthetic dataset? This issue matters to decide how significant the generalization is.
2. The method mentioned that there is a weight predicted in line-177. This weight prediction is important because the method interlacing between residual flow field and bundle adjustment, which is sensitive to outliers such as the ones caused by dynamic objects. However, there is no visualization or evaluation about this weighting in the paper. This issue also depends on the training data that if the training data are all static scenes how it can weigh properly on sequences with dynamic objects? So it would be good to have a clear discussion and analysis in the paper.

**Time Spent Reviewing:**

3.5

---

> ### Author Response · Authors · 2021-08-10
> **Thank you for your review. We address the individual points below.**
>
> **The paper does not mention the training data source.** We train exclusively on the synthetic TartanAir dataset (see Ln 260 which describes the training data source ). All results on real data demonstrate our network’s ability to generalize from real to synthetic data.
>
> **Missing visualization of the weights / importance of confidence weights.** Thank you for this  suggestion and we will add visualizations of confidence weights to the final version of our paper. We have tried training versions of our model where confidence weights are removed (replaced with all 1s). We have not been able to get stable training without confidence weights and generally found that this tends to produce singular matrices in the optimization layer. We are still working on pinpointing the issue, and will report our findings in the final version of our paper.
>
> **Limitation of Outlier Rejection.** Empirically, our method is robust on highly dynamic scenes, such as those present in the test sequences of TartanAir and ETH-3D. Dynamic objects are present in the training data we used (TartanAir), and the network relies on training to learn to reject outliers. This is possible because GRU maintains a history of the previous optimization steps, which makes it well equipped to reject outliers by means of the confidence map.

---

### Official Review · Reviewer_vRNK · 2021-07-20

**Rating:** 9
**Confidence:** 5

**Summary:**

This paper proposes a novel deep learning-based SLAM algorithm. The key technical contribution is to adopt a recurrently updated deep optical flow to serve as the estimation of correspondence and build a differentiable bundle adjustment solver module within the network. The proposed approach is solid and technically sound. It sets the new state-of-the-art and demonstrates exciting potential opportunities. The paper is clearly presented. I would strongly recommend accepting this paper.


**Main Review:**

Pros:
* Very strong results. Significantly better than prior art on TartanAir, EuRoC, ETH3D, and TUM and close to being real-time. It's exciting to see such results on SLAM.
* The approach is simple, elegant, and makes a lot of sense. I am particularly interested in the iterative learning of the residual flow.
* Less hyper-parameter tuning and engineering trick than most of the existing visual SLAM approaches.
* Easy to generalize to monocular depth, stereo, and RGB-d settings. It also demonstrates generalization capability across different datasets.


Cons/ Detailed Comments:


Writing:
* I would not claim the DBA layer is new — as you pointed out in the paper and some other existing literature, very similar instantiation has been seen before. It won’t hurt even if this part is not entirely novel — this submission has a good and novel design of the entire SLAM approach.
* There are not enough discussions to justify the technical choice and superior performance: 1) why rejection error is superior to the photometric ones as in indirect method; 2) why dense optical flow than sparse key feature correspondence? 3) why do you need the residual updates of the flow estimation?
* Do you see any failure cases in some testing videos? Could you show some failure cases?
* Qualitative results are not sufficient for this paper. I would like to see visualizations of the flow confidence map, intermediate flow & depth results, iterative updates, before vs after local BA, and an online demo video in the final supplementary.
*

Approaches:
* It’s not clear to me how did you select pixels for the local bundle adjustment (all or prune some by the confidence weight?) and how dense you store the features for global BA.
* Many papers have seen superior results using differentiable LM solver than GN solver, with a proper damping factor scheme. Have you compared both?
* Do you need a good initial pose estimation for a new incoming frame at the inference?
* Initialization from the first 12 frames is interesting to me: how do you initialize your pose then? All zeros? Won’t they fall into some local optimal? If not, do you have some intuitions why? This is different from the conventional approach, where we might need a good init from homography or essential decomposition.
* Have you double-checked whether Schur complement is indeed superior to other options, e.g., Cholesky in this case? Given the structure, Cholesky could be faster as well.
* It seems you directly remove the keyframe, have you considered conducting marginalization using the scour complement? I would love to see whether it makes a difference here.
* Is there any way for the network to decide on a tracking failure in some cases? This will be very useful in practice. Is the learned energy value a good indicator of success?


Experiments:
* There isn’t much ablation on the tech choice, e.g., optical flow-based results vs. only using dense depth-based results; different options for local BA windows.
* Following the above discussion on flow vs. feature matching, I am actually wondering how well superpoint+superglue + your proposed frontend DBA layer + your backend do. This would be a nice experiment.
* The scale of the global bundle adjustment is still very big. How do you deal with it? Do you directly implement with PyTorch tensors?
* Given the sparse block patterns, have you considered using a sparse tensor in the implementation to reduce memory footage and accelerate, like what has been done in ceres/gtsam?
* Is it straightforward to integrate a loop-closure module in the backend?


Relevant works:
* gradSLAM: Automagically differentiable SLAM, CVPR 2020: this is a similar end-to-end approach based on a differentiable bundle adjustment solver. That being said, both frontend and backend are different, and this submission’s result is stronger.
* Deep rigid instance scene flow, CVPR 2019; and RAFT-3D in CVPR 2020: these approaches have a similar DBA-layer optimizing reprojection error. The correspondence is based on deep flow estimation, but this approach is two-frame and mainly focuses on instance-level 3D scene flow. Besides, it would help if you discussed VOLSOR and Flowfusion more explicitly in related work, as they are also deep optical flow-based slam and directly relevant. This helps better position the paper and acknowledge prior work.

**Time Spent Reviewing:**

6

---

> ### Author Response · Authors · 2021-08-10
> **Thank you for your review. We address the individual points below.**
>
> **There are not enough discussions to justify the technical choice and superior performance.** Thank you for the suggestions, we will add more justification for our design choices.
>
> * **Why is reprojection error superior to photometric error?** Reprojection error is typically easier to optimize than photometric error. Photometric error functions are non-smooth since they rely on pixel-level Taylor approximations and typically require the use of image-pyramids to widen the basin of convergence. By using reprojection error, we inherit the smoother cost function and avoid the additional complexities of pyramid style inference.
>
> * **Why dense optical flow as opposed to sparse correspondence?** The main advantage of dense optical flow is increased robustness. Whereas sparse methods typically reason about each detected keypoints, we can leverage additional information just as lines and texture gradients. An additional advantage of dense flow is that we can condition the motion estimation of a given pixel on its neighbors which helps to resolve ambiguities such as repetitive textures.
>
> * **Why are residual flow updates necessary?** Interleaving flow updates and optimization is a central piece of our architecture. This gives the network greater control of the optimization process, giving it a way to correct residuals, down-weight outliers, and use the history encoded in the GRU state. In one of our ablations, (Fig. 6, right) we show that directly predicting flow instead of interleaving flow updates and bundle adjustment leads to much worse performance.
>
> **Qualitative results are not sufficient (e.g visualizations of confidence weights, optical flow, depth maps, demo video).** This is a good suggestion, and we will add these visualizations to the final version of our paper in addition to creating a video demonstrating our system.
>
> **How are pixels selected for local/global bundle adjustment?** Every pixel of every keyframe is used for bundle adjustment. While the network can downweight certain flow estimates, no pixel is outright excluded.
>
> **Many papers have seen superior results using differentiable LM solver than GN solver, with a proper damping factor scheme. Have you compared both?** We have not tried any differentiable damping factor schemes. Typically, damping factors are useful for improving the basin of convergence, but we have not observed any convergence issues with our BA layer.
>
> **Do you need a good initial pose estimation for a new incoming frame at the inference?** We did not find the pose initialization to make much difference. We tried two methods for initialization: (1) initializing the new frame with the pose of the previous frame and (2) using a constant velocity motion model. Both gave similar accuracy.
>
> **Initialization from the first 12 frames is interesting to me: how do you initialize your pose then?** We initialize all poses to the identity transformation.  In the classical SLAM setting, this can result in failure to converge. However, since our network interleaves flow updates and Bundle Adjustment, we find that the network can guide the optimization process to the global minimum. On all the sequences we tested (over 100), we did not observe any cases where the network failed to initialize. That said, there certainly are cases where this simple method won’t work, like trying to initialize on a purely planar scene. In these cases, more sophisticated algorithms need to be used.
>
> **Have you double-checked whether Schur complement is indeed superior to other options, e.g., Cholesky in this case?** Although we use the Schur complement, we still use Cholesky factorization on the reduced camera block to solve for the camera pose update. We also tried directly performing sparse Cholesky factorization on the full system using the CHOLMOD library, but found this to be much slower than directly implementing the Schur complement. An additional advantage of using the Schur complement is that it can be easily parallelized on the GPU.
>
> **It seems you directly remove the keyframe, have you considered conducting marginalization instead?** This is possible with our framework, but not something we have had a chance to try yet.
>
> **There isn’t much ablation on the tech choice, e.g., optical flow-based results vs. only using dense depth-based results; different options for local BA windows.** Thank you for this suggestion. We will add additional ablations in the final version of our paper.
>
> **Following the above discussion on flow vs. feature matching, I am actually wondering how well superpoint+superglue + your proposed frontend DBA layer + your backend do. This would be a nice experiment.**
> Thank you for this suggestion. We have not been able to complete this experiment as of writing, but we hope to include it in the final version of our paper. It is worth noting that the winner of the ECCV 2020 SLAM competition on TartanAir is based on superpoint + superglue + global BA (COLMAP), and is outperformed by our system (table 2).
>
> **The scale of the global bundle adjustment is still very big. How do you deal with it? Do you directly use PyTorch tensors?** During training, we implement the bundle adjustment layer directly in PyTorch using dense tensor operations in order to take advantage of the PyTorch automatic differentiation library. All operations can be fully vectorized leading to reasonable performance on small problems.
>
> At inference time, we use a custom CUDA implementation, which leverages the block sparse structure of the problem. We put a fair amount of effort into optimizing our BA implementation. While the number of variables is large, Bundle Adjustment can be parallelized on the GPU. Each iteration for 7 frames and 30 dense flow fields takes 1.2 ms. For 200 keyframes frames / 1000 dense flow fields, it takes 12 ms.
>
> **Given the sparse block patterns, have you considered using a sparse tensor in the implementation to reduce memory footage and accelerate, like what has been done in ceres/gtsam?** Yes, we implement our own CUDA block sparse operations and use the CHOLMOD library to perform Cholesky factorization on the reduced camera block. At this time, ceres and gtsam do not yet support GPU acceleration.
>
> **Is there any way for the network to decide on a tracking failure in some cases? This will be very useful in practice. Is the learned energy value a good indicator of success?** We found the learned energy following BA (residual magnitude in Eq. 4) to be a good indicator of success. We induced failure cases in our method by creating 10-20 frame gaps in the video and looked at the resulting residuals. By using a simple threshold, we were able to detect most failures.
>
> **Is it straightforward to integrate a loop-closure module in the backend?** Yes, loop closure is already performed in the backend. We could further extend the work to perform visual relocalization, but this is not implemented in the current system.
>
> **Related Works:** We will add a discussion of the works gradSLAM, VOLSOR, and FlowFusion.

---

### Decision · Program_Chairs · 2021-09-27

**Decision:**

Accept (Oral)

**Comment:**

The paper received strong reviews and I am happy to recommend it for acceptance. I encourage the authors to make the improvements that they indicated in their rebuttal in the final version.